# C-Reactive Protein Monitoring and Clinical Presentation of Fever as Predictive Factors of Prolonged Febrile Neutropenia and Blood Culture Positivity after Autologous Hematopoietic Stem Cell Transplantation—Single-Center Real-Life Experience

**DOI:** 10.3390/jcm11020312

**Published:** 2022-01-09

**Authors:** Daniela Carcò, Uros Markovic, Paolo Castorina, Valeria Iachelli, Tecla Pace, Paola Guardo, Gabriella Amato, Federica Galbo, Paola Scirè, Gaetano Moschetti

**Affiliations:** 1Unità Operativa di Laboratorio Analisi Chimco Cliniche, Mediterranean Institute of Oncology, Via Penninazzo 7, 95029 Viagrande, Italy; daniela.carco@grupposamed.com (D.C.); castorina@ct.infn.it (P.C.); valeria.iachelli@grupposamed.com (V.I.); pace.tecla@grupposamed.com (T.P.); paola.guardo@grupposamed.com (P.G.); 2Unità Operativa di Oncoematologia e BMT Unit, Mediterranean Institute of Oncology, Via Penninazzo 7, 95029 Viagrande, Italy; gabriella.amato@grupposamed.com (G.A.); federica.galbo@grupposamed.com (F.G.); paola.scire@grupposamed.com (P.S.); gaetano.moschetti@grupposamed.com (G.M.); 3Department of Biomedical, Dental, Morphological and Functional Imaging Sciences, University of Messina, 98122 Messina, Italy; 4Dipartimento di Fisica, Istituto Nazionale di Fisica Nucleare, Sezione di Catania, 95123 Catania, Italy; 5Faculty of Mathematics and Physics, Charles University, V Holešovičkach 2, 18000 Prague, Czech Republic; 6Dipartimento di Fisica e Astronomia, Università di Catania, 95123 Catania, Italy

**Keywords:** autologous stem cell transplantation, febrile neutropenia, C-reactive protein, blood culture positivity

## Abstract

Background: Febrile neutropenia (FN) is a medical emergency that requires urgent evaluation, timely administration of empiric broad-spectrum antibiotics and careful monitoring in order to optimize the patient’s outcome, especially in the setting of both allogeneic and autologous hematopoietic stem cell transplant (ASCT). Methods: In this real-life retrospective study, a total of 49 consecutive episodes of FN were evaluated in 40 adult patients affected by either multiple myeloma (thirty-eight) or lymphoma (eleven), following ASCT, with nine patients having fever in both of the tandem transplantations. Results: Febrile neutropenia occurred a median of 7 days from ASCT. Median duration of FN was 2 days, with 25% of population that had fever for at least four days. Ten patients had at least one fever spike superior to 39 °C, while the median number of daily fever spikes was two. Twenty patients had positive blood cultures with XDR germs, namely *Pseudomonas aeruginosa* and *Klebsiella pneumoniae*, present in seven cases. ROC analysis of peak C-reactive protein (CRP) values was conducted based on blood culture positivity and a value of 12 mg/dL resulted significant. Onset of prolonged fever with a duration greater than 3 days was associated with the presence of both a peak number of three or more daily fever spikes (*p* = 0.02) and a body temperature greater than 39 °C (*p* = 0.04) based on odds ratio (OR). Blood culture positivity and peak CRP values greater than 12 mg/dL were also associated with prolonged fever duration, *p* = 0.04, and *p* = 0.03, respectively. The probability of blood culture positivity was also greater in association with fever greater than 39 °C (*p* = 0.04). Furthermore, peak CRP values below the cut-off showed less probability of positive blood culture (*p* = 0.02). Conclusions: In our study, clinical characteristics of fever along with peak CRP levels were associated with a higher probability of both prolonged fever duration and positive blood culture, needing extended antibiotic therapy.

## 1. Introduction

Febrile neutropenia (FN) is a medical emergency that requires urgent evaluation, timely administration of empiric broad-spectrum antibiotics and careful monitoring in order to optimize the patient’s outcome and mitigate the risk of complications, especially in the setting of hematopoietic stem cell transplant, both autologous (ASCT) and allogeneic. If not treated promptly, it represents a major cause of ASCT-related morbidity and mortality due to high dose chemotherapy and prolonged engraftment [1]. In this respect, a possible role of C-reactive protein (CRP) in early risk assessment prior to and in the course of FN was already evaluated [2,3,4]. Additionally, microbiological study of the patient including surveillance swab tests, especially rectal, sputum, urine and stool culture analysis could help guide antibiotic management, anticipating eventual positive blood culture results [5,6,7]. The value of the abovementioned examinations together with clinical presentation of the fever itself could aid in FN severity evaluation, with eventual promptly initiation of targeted antimicrobial therapy, thus improving fever outcome in the extremely fragile subset of patients with severe immunosuppression [8,9].

In this retrospective study we discuss clinical presentation of the fever itself and CRP monitoring, and their role as possible predictive factors to estimate FN severity, in terms of duration and blood culture positivity in ASCT setting of multiple myeloma and lymphoma patients. 

## 2. Materials and Methods

### 2.1. Patient Selection

In this real-life retrospective study, a total of 49 consecutive episodes of FN were evaluated in 40 patients ≥ 18 years old (9 patients had an FN episode in both transplantations) affected by multiple myeloma or lymphoma who received ASCT between January 2018 and December 2020 in our center. Data from each patient were collected from electronic medical records from the time of admission until discharge. During the hospitalization, all patients remained in an isolation unit with high-efficiency particulate air (HEPA) filtration and had a double lumen central venous catheter (CVC) positioned in the internal jugular vein. The patient’s median age at admission was 58 years (range 21–70 years), with 35 male and 14 female patients. Karnofsky performance status (PS) at admission was 80% or better in 42 hospitalizations, while inferior PS was present in 7 cases. As mentioned previously, nine patients were evaluated twice given that the episodes of fever were detected in both of their tandem ASCT, as part of the scheduled program. Patients’ characteristics are described in Table 1. The study was conducted in accordance with International Conference on Harmonization Guidelines on Good Clinical Practice and the principles of the Declaration of Helsinki and all patients provided written informed consent to data recording and collection prior to ASCT. 

The primary endpoint was to evaluate the impact of CRP monitoring and clinical presentation of fever (day of fever onset according to ASCT reinfusion date, daily peak number of fever spike episodes with a body temperature greater than 38 °C, occurrence of a fever spike greater than 39 °C) on FN severity, in terms of fever duration, and blood culture positivity. Secondary endpoints were the assessment of CRP and white blood cell (WBC) monitoring as early predictive factors of fever onset, along with the evaluation of patient (age, PS status, previous infectious episodes), disease (diagnosis, response status prior to ASCT according to American Society for Blood and Marrow Transplantation (ASBMT), ASCT (previous transplantations, conditioning regimen, number of viable CD34+ cells reinfused, number of days from reinfusion until engraftment) and microbiological characteristics (baseline swab microbiology for multi-drug resistant germs and multi-drug resistant germ bacteremia) in terms of FN severity and blood culture positivity.

### 2.2. Conditioning Regimen and Supportive Care

All patients received high-dose myeloablative chemotherapy as part of the conditioning regimen. High doses of melphalan IV were used in multiple myeloma (MM) patients two days prior to ASCT, while a 7-day FEAM conditioning regimen, including fotemustine, etoposide, cytarabine and melphalan IV was used in all but one lymphoma of the patients, who received high doses of cytarabine. All patients received antiviral prophylaxis treatment with acyclovir 800 mg orally or 500 mg intravenous twice daily, and antifungal prophylaxis with fluconazole 400 mg intravenous daily from admission, according to the center’s policy. Thirty-four patients received levofloxacin as antibiotic prophylaxis from day 2 after autologous stem cell reinfusion, while ceftriaxone was used subsequently in fifteen hospitalizations, according to local antimicrobial resistance. In case of two or more episodes of diarrhea, stool culture was requested, and metronidazole antibiotic therapy was started at standard dose. Daily granulocyte colony-stimulating factor (G-CSF) treatment with lenograstim, 263 mcg, was started from day 3 after reinfusion in order to enhance stem cell engraftment and minimize the morbidity and mortality associated with prolonged neutropenia, according to ASCO guidelines [10] and the center’s policy. Engraftment was defined as ANC greater than 0.5 × 10^9^/L for three consecutive days following ASCT, with subsequent G-CSF suspension.

### 2.3. Screening and Management of Febrile Neutropenia 

Serum CRP levels were measured regularly on a daily basis during transplant, independently from body temperature. Patients were evaluated for possible colonization at admission with oropharyngeal, nasal, axillary, inguinal, vaginal and rectal (including multi-drug resistant pathogens) microbiological swab tests, sputum, urine and stool culture analysis. Neutropenia was defined as an absolute neutrophil count (ANC) < 0.5 × 10^9^/L or ANC count < 1.0 × 10^9^/L that was expected to fall below 0.5 × 10^9^/L in the following 48–72 h, while fever was defined as a single axillary temperature of >38.3 °C or at least 38 °C sustained over one-hour period [11,12].

In the event of FN, aerobic, anaerobic and fungal blood cultures with antibiotic susceptibility test were obtained from both CVC and peripheral veins, with additional aerobic and anaerobic peripheral venous blood cultures 30 min after an episode of fever. In case of isolated pathogens with specific antibiogram from baseline microbiological screening, targeted IV antibiotic therapy was initiated. On the other hand, in the absence of resistant pathogens antibiotic prophylaxis was substituted with broad-spectrum intravenous antibiotics, like piperacillin-tazobactam 4.5 g three times a day or ampicillin-sulbactam 3 g four times a day, followed by amikacin 500 mg twice daily. The antibiotic and antifungal administration pattern was recorded for each FN episode. Temperature measurements were performed at least three times a day and, in the case of persistent fever, blood cultures were repeated at least once every 48 h from both CVC and peripheral vein. Additional imaging and microbiological tests, such as urine and stool culture, were requested based on the patient’s clinical status. A physical check-up was performed daily reporting eventual clinical evidence associated with FN, such as respiratory symptoms or catheter inflammation (Table 1).

### 2.4. Statistical Analysis 

Continuous variables were summarized using the median and range (minimum and maximum). Categorical variables were summarized in counts and percentages. Fever duration was evaluated from the day of first fever spike until the day of fever resolution. Fever duration of more than three days was chosen arbitrarily given that, in case of drug-resistant germs, initial empirical antibiotic therapy has little effect on the FN outcome in the first 48–72 h. Additionally, age, peak number of daily fever spikes, body temperature greater than 39 °C and day of fever onset from autologous CD34+ cell infusion were also chosen arbitrarily, based on clinical experience of fever episodes in the ASCT setting. Number of CD34+ cells reinfused was evaluated as predictive factor due to its impact on neutrophil engraftment. 

Receiver operating characteristic (ROC) analysis was utilized to evaluate initial values of both WBC and CRP at fever onset in 39 patients, due to lack of data availability in the rest of study population, in order to obtain specific cut-off values. Combined study of sensitivity and specificity with Youden index was utilized in the analysis of WBC and CRP, both singularly and in approximate binormal ROC curve, based on the presence of a body temperature greater than 39 °C, a peak number of daily fever spikes and duration of fever.

On the other hand, peak value of CRP in course of FN was evaluated in terms of fever duration and blood culture positivity in the whole cohort (49 episodes of febrile neutropenia). A subset of potential predictive factors was evaluated singularly with both fever duration and blood culture positivity by using the odds ratio (OR) and 95% confidence intervals (95% CIs) of. A two-tailed *p* value < 0.05 was considered to be statistically significant. Furthermore, each of the factors in the final model was fitted into multi-variable logistic regression in order to estimate the unadjusted univariate ORs.

All calculations were performed using MedCalc version 12.30.0.0 (Producer: MedCalc Software bvba, Ostend, Belgium), www.medcalc.org (accessed on 18 June 2021).

## 3. Results

### 3.1. Disease Baseline Characteristics

In the study cohort of 49 evaluated patients with FN following ASCT, thirty-eight patients were affected by multiple myeloma, while eleven had lymphoma, namely four Hodgkin lymphoma, two follicular, two diffuse large B cell lymphoma, while mantle cell, primary mediastinal B-cell and primary cerebral non-Hodgkin lymphoma were present in one patient each. Thirty-two patients underwent ASCT as part of the first treatment line with nine undergoing double ASCT as programmed. Transplantation was performed in the second and third line of therapy in six and two patients, respectively. The patients were classified according to ASBMT disease classification and treatment response prior to ASCT that can be found in Table 1. Twelve patients had previous infective episodes prior to transplantation, namely six cases of pneumonia, meningitis and sepsis in two each and a one case of viral myocarditis and syphilis, respectively. All patients underwent induction or salvage chemotherapy prior to ASCT and 16 out of 24 patients (67%) with multiple myeloma had IgM values inferior to 40 mg/dL. Immunoglobulin G values were not evaluated, given that the majority of MM patients had monoclonal IgG protein.

### 3.2. Autologous Hematopoietic Stem Cell Transplantation

A high dose chemotherapy conditioning regimen was chosen according to the patient’s baseline disease. Melphalan was utilized as a myeloablative conditioning regimen in twenty-nine MM patients that underwent a total of 38 ASCT procedures, with nine patients that had FN in both transplantations. A myeloablative melphalan dosage of 200 mg/m^2^ was used in half of MM patients, while ulterior 19 cases were treated at reduced dosage between 100 and 150 mg/m^2^, based on advanced patient’s age and comorbidities. On the other hand, ten lymphoma patients underwent a myeloablative FEAM conditioning regimen at pre-established drug doses, while in the case of primary cerebral lymphoma high doses of cytarabine were used. A median number of 4.2 × 10^6^/kg viable CD34+ cells mobilized from peripheral blood was reinfused (range 2–14.8 × 10^6^/kg), with each patient receiving a minimum of 2 × 10^6^/kg of CD34+ cells.

Engraftment was evidenced at a median of 11 days (range 9–16 days) following PBSC reinfusion, with nearly 90% of the population achieving it within 13 days from ASCT.

### 3.3. Microbiological Status Prior to ASCT

Patients underwent microbiological screening at admission, including for multi-drug-resistant pathogens as previously mentioned. Gram positive *Staphylococcus* were present in at least one swab isolation in the whole cohort, with the most frequent germs in around half of the populations being *Staphylococcus* (St.) epidermidis, *St. aureus* and *St. haemolyticus*. Other Gram-positive bacteria isolated in a limited number of patients (three or less) were following: *Enterococcus faecalis*, *St. hominis*, *St. lugdunensis*, *Streptococcus* (S.) *pyogenes*, *St. saprophyticus* and *St. xylosus*. The most frequently isolated Gram-negative non-multidrug-resistant bacteria in the study population was *Escherichia coli* in 15 patients, with extended-spectrum β-lactamase in most cases, while *Stenotrophomonas maltophilia*, proteus mirabilis and enterobacter cloacae were isolated in one patient each. A total of six patients returned a positive swab test for yeast, *Candida albicans* in three patients, *Candida parapsilosis* and *Candida krusei* in one patient each, while one patient had both *Candida krusei* and *Candida glabrata*. As for the extremely drug-resistant (XDR) germs, carbapenemase-producing *Pseudomonas aeruginosa* and *Klebsiella pneumoniae* (KPC) were isolated in a total of 14 patients (twelve and two patients, respectively), mainly from a rectal swab at admission. 

### 3.4. Characteristics of Febrile Neutropenia and Antimicrobial Treatment

Febrile neutropenia occurred at a median of 7 days from PBSC reinfusion (range 2–11 days), associated with chills in all patients. Twelve patients suffered early fever onset within the first 5 days from ASCT. Median duration of FN was 2 days (range 1–9 days), with 25% of study population having fever for four or more days. Ten patients had at least one fever spike superior to 39 °C, while the median peak number of daily fever spikes was 2 (range 1–5). Signs and symptoms associated with FN along with other clinical characteristics of the fever episode are described in Table 2. One patient passed away after developing septic shock with a positive blood culture for *Pseudomonas aeruginosa* XDR and extensive pneumonia. The patient was treated with both empiric and targeted antibiotics and antifungal therapy, including ceftolozane/tazobactam and isavuconazole, but eventually passed away seven days from FN onset.

Twenty patients had positive blood cultures with one patient having more than one positive culture. Half of the patients had *Staphylococcus* spp., while XDR germs were isolated in seven patients, respectively, and *Pseudomonas aeruginosa* and KPC in five and two patients. The site of blood culture isolation is described in Table 2. Three patients suffered from sepsis according to SOFA score [13] with subsequent septic shock, and one fatal outcome.

As for the empirical antibiotic treatment following FN, prophylaxis with levofloxacin or ceftriaxone was suspended and piperacillin/tazobactam or ampicillin/sulbactam was started in around 90% of the patients. One third of the patients were treated with empirical therapy alone, including amikacin, with fever resolution. In case of persistent fever, patients with suspected Gram-positive germs daptomycin, linezolid and/or tygacil were initiated based on possible infection site. Carbapenems were used when Gram-negative germs were suspected, namely imipenem and meropenem. One third of patients initiated novel antibiotic treatment with ceftozolane/tazobactam based on admission isolation of XDR germs or persistently high fever without benefit from prior antibiotic treatment (Table 3). Antifungal treatment with caspofungin and isavuconazole was used in two patients each, due to suspected fungal infection.

### 3.5. Monitoring of CRP and WBC on Febrile Neutropenia Onset

Daily values of WBC, CRP and body temperature were recorded following ASCT in 580 days of 39 FN episodes, verifying fever onset and eventual sepsis. The analysis started the second day after ASCT (541 data days). The complete group of patients had no fever on 483/541 days (89.3%) and the onset of fever on 58/541 days (11.7%), with no sepsis on 36/58 days and with sepsis on 22/58 days, given that sepsis can be diagnosed after fever onset. In order to verify the possible role of WBC and/or CRP cut-off values in predicting fever onset, analysis of ROC curves of the 541 daily results was carried out. In particular, possible threshold values of WBC or CRP the day before the detection of fever were studied. The ROC curve for WBC observed the day before fever showed an area under curve (AUC) around 92%. The combined study of sensitivity and specificity (Youden index) gave a cut-off of less than 300 WBC for the onset of fever the day after. An analogous ROC curve for CRP measured the day before fever showed weak impact on fever onset, given that AUC resulted around 60%.

Moreover, WBC and CPR levels the day before the onset of FN were analyzed in an approximate binormal ROC analysis for eventual statistical correlation with high fever (>39 °C), the peak number of daily fever spikes and fever duration. The maximum likelihood estimation of the binormal ROC curve reported no correlation. Indeed, the ROC curve for peak number of daily fever spikes showed an AUC of about 50% both for WBC and CPR. Analogous results were obtained for the peak number of fever spikes, while only a weak signal of a cut-off correlation was evidenced for the duration of fever (>3 days) in terms of CPR with AUC ≃ 0.64.

### 3.6. Predictive Factors of FN Duration and/or Blood Culture Positivity

In an attempt to identify possible predictive factors, which could be of aid in the evaluation of FN non-responsive to first-line antibiotics and eventual fever from bacteremia with positive blood culture, OR analysis was used. Additionally, ROC analysis of peak CRP values was conducted based on blood culture positivity and a value of 12 mg/dL resulted as significant. As already mentioned, patient, disease, ASCT, microbiological and fever characteristics were assessed (Table 4).

Fever duration greater than 3 days was associated with the presence of both a peak number of 3 or more daily fever spikes (OR 5.54, 95% CI 1.27–24.1, *p* = 0.02) and body temperature greater than 39 °C (OR 4.57, 95% CI 1.03–20.18, *p* = 0.04). Blood culture positivity and peak CRP values greater than 12 mg/dL were also associated with prolonged fever duration, OR 4.17, 95% CI 1.04–16.62, *p* = 0.04 and OR 5.88, 95% CI 1.13–30.63, *p* = 0.03, respectively. The probability of blood culture positivity was also greater in association with clinical presentation of fever, mainly with fever greater than 39 °C (OR 4.67, 95% CI 1.03–21.07, *p* = 0.04) and prolonged fever duration as mentioned above. Furthermore, patients with peak CRP values below the cut-off had a smaller probability of having positive blood culture (OR 4.25, 95% CI 1.21–14.88, *p* = 0.02).

Multiple logistic regression analysis was conducted in order to analyze the predictive factors together, both for prolonged fever duration and blood culture positivity, and failed to confirm their impact.

## 4. Discussion

Febrile neutropenia mortality in hematological malignancies is extremely frequent, with a particularly high risk of mortality in patients treated with high dose chemotherapy for acute leukemia and in the case of hematopoietic stem cell transplantation [14]. Due to prolonged neutropenia, prompt initiation of broad-spectrum empiric antimicrobial therapy at first fever episode has become the standard of care for potential infections due to blood culture positivity [11,15]. Even so, in the era of multidrug resistant bacteria, the management of FN is becoming a growing issue. History of prior colonization and/or bloodstream infection caused by XDR Gram-negative bacteria resistant to empiric antibiotic treatment, such as *Klebsiella* spp., *Pseudomonas aeruginosa*, *Escherichia coli*, etc. represents an important diagnostic tool in neutropenic patients [16]. In this specific subset of patients, anticipation of targeted antibiotic therapy could overcome the limits of classical empiric antibiotics, preventing treatment delay and improving FN outcome in case of multi-drug resistant bacteria [17,18]. However, not all patients have a history of prior isolation of an antimicrobial-resistant strain of bacteria and, in the case of poor response to empiric broad-spectrum antibiotics, waiting for blood culture results that are still undergoing analysis, there is urgent need for both clinical and biochemical parameters that could help in FN management.

In our study, we analyzed a total of 49 FN episodes following ASCT. Fourteen patients showed colonization of XDR Gram-negative germs, namely *Pseudomonas aeruginosa* and KPC at admission, while, of 20 patients with positive blood culture, XDR germs were isolated in one third (Table 2). As for the antibiotic treatment, two-thirds of the population needed ulterior antibiotics, following empiric therapy, with one third of the patients receiving novel ceftolozane/tazobactam antibiotic, while in six patients fosfomycin and/or colomycin were used (Table 3).

The study cohort was then divided in terms of patient, disease, ASCT, fever, biochemical and microbiological characteristics in order to evaluate possible predictive value in terms of FN severity, mainly duration, and blood culture positivity in advance. Clinical characteristics of fever represent an important tool in febrile neutropenia evaluation, especially after the initiation of empiric antibiotics. According to the experience in our center, the frequency of peak daily fever spikes and body temperature greater than 39 °C represent important signs of FN severity and were assessed as such. The OR analysis confirmed the importance of 39 °C body temperature in terms of both prolonged FN duration and blood culture positivity, while three or more peak daily fever spikes were predictive of prolonged fever duration alone. Prolonged fever duration was also predictive in patients with positive blood cultures. Peak values of CRP were also assessed, first with ROC analysis, confirming a cut-off value of 12 mg/dL, and its role was verified in both prolonged FN and blood culture positivity.

The role of WBC and CRP in predicting the onset of FN was also assessed. In particular, by daily monitoring of WBC and CRP after ASCT for our sample of 39 patients for 580 days, certain cut-off values of WBC or CRP could signal the possibility of fever onset one day in advance. The suggested criterion could be useful in the clinical practice, given the fragility of the ASCT population. However, the cut-off was not able to recognize sepsis prematurely, which is among the leading causes of death in intensive care units, due to the inadequacy of CRP and white blood cell count alone in predicting bacteremia of patients with suspected infection [19,20]. Indeed, the advent of an affluence of available digital health data has created a setting in which machine learning can be used for early prediction of sepsis, on the basis of quite a large set of vital parameters [21,22,23]. According to our sample, a WBC cut-off of 300/mmc gave a strong indication of the onset of FN the day after. On the other hand, the analogous analysis for the CPR level had weak indication. No statistical cut-off correlation of the approximate binormal ROC analysis has been observed for fever characteristics, such as high temperature (>39 °C), peak number of daily fever spikes and duration of febrile condition, in terms of WBC and CPR. In our experience, the CRP values at FN onset are subsequently influenced by the sensibility to initial treatment response to empiric antibiotics. Therefore, while empiric antibiotics and later changes in antibiotic approach were based on fever persistence, routine CRP monitoring represented a valid aid in evaluating treatment response together with associated fever characteristics.

The study was retrospectively designed with limitations, in terms of population size and heterogeneity, with gender and disease predominance, and nine patients were evaluated twice given the double ASCT.

## 5. Conclusions

In our study, clinical characteristics of fever and peak CRP levels were associated with higher probability of both prolonged fever duration and positive blood culture, needing extended antibiotic therapy. Based on our experience, routine daily CRP measurements together with the evaluation of fever characteristics could aid in the management of FN in the SCT setting. Further studies are needed in order to assess the importance of clinical presentation together with CRP and other infectious biomarkers, thus improving the outcome of patients affected by febrile neutropenia following ASCT.

## Figures and Tables

**Table 1 jcm-11-00312-t001:** Clinical characteristics in 40 multiple myeloma/lymphoma patients (total 49 episodes) of febrile neutropenia following ASCT.

**Age (Total 49 Episodes)**	
Median in years (range)	58 (21–70)
≤65 years, N (%)	37 (75)
>65 years, N (%)	12 (25)
**Gender (total 49 episodes)**	
Male, N (%)	35 (71)
Female, N (%)	14 (29)
**Hematological malignancy (total 49 episodes)**	
Multiple myeloma, N (%)	38 (77)
Lymphoma, N (%)	11 (23)
**Treatment line (total 49 episodes)**	
First line, N (%)	41 (84)
Second line or higher, N (%)	8 (16)
**Autologous stem cell transplantation (total 49 episodes)**	
First, N (%)	33 (67)
Second, N (%)	16 (33)
**Treatment response prior to ASCT (total 49 episodes)**	
≥VGPR, N (%)	26 (53)
≤PR, N (%)	23 (47)
**Prior infectious episodes (total 49 episodes)**	
No episodes, N (%)	37 (75)
One or more episodes, N (%)	12 (25)
**ASBMT disease classification (total 49 episodes)**	
Low, N (%)	40 (82)
High, N (%)	9 (18)

Abbreviations: ASCT—autologous stem cell transplantation; VGPR—very good partial response; PR—partial response; ASBMT—American Society for Blood and Marrow Transplantation.

**Table 2 jcm-11-00312-t002:** Fever characteristics in 49 episodes of febrile neutropenia.

**Fever Onset from ASCT**	
Median in days (range)	7 (2–11)
≤5 days, N (%)	11 (22)
≥6 days, N (%)	38 (78)
**Fever duration**	
Median in days (range)	2 (1–9)
≤3 days, N (%)	37 (75)
>3 days, N (%)	12 (25)
**Fever ≥ 39 °C**	
Yes, N (%)	10 (20)
No, N (%)	39 (80)
**Signs and symptoms associated to FN**	
CVC local inflammation, N (%)	4 (8)
Dyspnea, N (%)	3 (6)
Skin rash, N (%)	1 (2)
Dysuria, N (%)	1 (2)
**Peak number of daily fever spikes**	
Median (range)	2 (1–5)
<3 fever spikes, N (%)	27 (55)
≥3 fever spikes, N (%)	22 (45)
**Blood cultures**	
Negative, N (%)	29 (59)
Positive, N (%)	20 (41)
**Site of positive blood culture (20 patients)**	
CVC, N (%)	7 (35)
PV, N (%)	4 (20)
CVC + PV, N (%)	9 (45)
**Blood culture isolations (20 patients)**	
*Staphylococcus* spp., N (%)	10 (50)
*Pseudomonas aeruginosa* XDR, N (%)	5 (25)
*Klebsiella pneumoniae* XDR, N (%)	2 (10)
*Escherichia coli* ESBL, N (%)	2 (10)
*Stenotrophomonas maltophilia*, N (%)	1 (5)

Abbreviations: ASCT—autologous stem cell transplantation; FN—febrile neutropenia; CVC—central venous catheter; PV—peripheral vein; XDR—extremely drug resistant; ESBL—extended spectrum beta-lactamase.

**Table 3 jcm-11-00312-t003:** Antibiotic and antifungal treatment in 49 episodes of febrile neutropenia.

**Antibiotic Therapy**	
Empirical antibiotics only ^!^, N (%)	16 (33)
Gram-positive antibiotics *, N (%)	26 (53)
Ceftolozane/tazobactam	16 (33)
Carbapenems ^#^, N (%)	9 (18)
Fosfomycin and/or Colomycin	6 (12)
**Antifungal therapy**	
Caspofungin, N (%)	2 (5)
Isavuconazole, N (%)	2 (5)

^!^ piperacillin/tazobactam, amikacin and ampicillin/sulbactam; * daptomycin, linezolid and tygacil; ^#^ meropenem and imipenem.

**Table 4 jcm-11-00312-t004:** Odds ratio analysis for fever duration and blood culture positivity.

		Fever Duration > 3 Days				Blood Culture Positivity			
CATEGORIES		No	Yes	Odds Ratio (95% CI)	*p*-Value	No	Yes	Odds Ratio (95% CI)	*p*-Value
**Age**	≤65 years	28	9	1.04 (0.23–4.68)	0.96	22	7	1.05 (0.28–3.93)	0.95
	>65 years	9	3			15	5		
**Disease**	Multiple myeloma	30	7	2.14 (0.5–9.18)	0.3	24	5	2.06 (0.53–7.99)	0.3
	Lymphoma	8	4			14	6		
**Karnofsky PS**	≥80%	33	4	2.75 (0.52–14.59)	0.23	27	2	4.5 (0.78–26.08)	0.09
	<80%	9	3			15	5		
**ASBMT disease classification**	Low	31	6	1.72 (0.36–8.29)	0.5	24	5	1.2 (0.28–5.16)	0.8
	High	9	3			16	4		
**ASCT**	First	24	13	0.61 (0.14–2.68)	0.52	20	9	0.77 (0.36–4.01)	0.77
	Second	9	3			13	7		
**Prior infectious episodes**	No	28	9	1.04 (0.23–4.68)	0.96	19	10	0.21 (0.04–1.1)	0.06
	Yes	9	3			18	2		
**Conditioning regimen**	Reduced dose	13	24	0.54 (0.15–2.02)	0.36	13	16	1.89 (0.57–6.32)	0.29
	Standard dose	6	6			6	14		
**Reinfused CD34+ cells**	≥4 × 10^6^/kg	14	23	0.44 (0.12–1.64)	0.22	15	14	2.5 (0.75–8.31)	0.13
	<4 × 10^6^/kg	7	5			6	14		
**XDR germ * swab positivity**	No	26	11	0.79 (0.18–3.48)	0.75	21	8	1.12 (0.32–3.95)	0.85
	Yes	9	3			14	6		
**Day of fever onset from ASCT reinfusion**	>5 days	31	6	3.69 (0.87–15.62)	0.08	23	6	1.28 (0.33–4.94)	0.73
	≤5 days	7	5			15	5		
**Fever duration**	>3 days	//	//	//	//	25	4	4.17 (1.04–16.62)	** *0.04* **
	≤3 days	//	//			12	8		
**Fever spike ≥ 39 °C**	No	32	5	4.57 (1.03–20.18)	** *0.04* **	26	3	4.67 (1.03–21.07)	** *0.04* **
	Yes	7	5			13	7		
**Peak n. of daily fever spikes**	<3	24	13	5.54 (1.27–24.1)	** *0.02* **	18	11	2 (0.63–6.36)	0.24
	≥3	3	9			9	11		
**XDR germ * blood culture positivity**	No	8	4	0.67 (0.09–4.93)	0.69	N.E.	N.E.	N.E.	N.E.
	Yes	6	2			N.E.	N.E.		
**CRP level >12 mg/dL**	No	20	17	5.88 (1.13–30.63)	** *0.03* **	17	12	4.25 (1.21–14.88)	** *0.02* **
	Yes	2	10			5	15		

* Carbapenemase producing *Klebsiella pneumoniae* and Pseudomonas aeruginosa; Abbreviations: PS—performance status; XDR—extremely drug resistant; ASCT—autologous stem cell transplantation; CRP—C reactive protein; CI—confidence interval; N.E.—not evaluable. Italics are used to underline the statistical significans.

## Data Availability

The data that support the findings of this study are available from the corresponding author, U.M., upon reasonable request.

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
