# Peer review of "C-Reactive Protein Monitoring and Clinical Presentation of Fever as Predictive Factors of Prolonged Febrile Neutropenia and Blood Culture Positivity after Autologous Hematopoietic Stem Cell Transplantation—Single-Center Real-Life Experience"

_jcm, 2022, doi:10.3390/jcm11020312_

Round 1

Reviewer 1 Report

In this paper the Authors reported as fever along with peak and CRP levels >12 after autologous hematopoietic stem cell transplantation were associated with higher probability of both prolonged fever duration and positive blood culture, needing extended antibiotic therapy. Thus the close monitoring of CRP and the evaluation of clinical presentation of fever are important infectious biomarkers that may help the clinical approach of this setting of immunosoppressive patients.

Author Response

Dear reviewer we thank You for the evaluation and comments provided.

Reviewer 2 Report

This is a well written and statistically sound paper that investigated the use of CRP and clinical characteristics as predictive parameters in prolonged febrile neutropenia in transplant patients. The reviewer only has some minor comments/concerns:

  • was CRP measured daily? The authors show that the CRP  level of 12 mg/dL is predictive of prolonged FN, however there was no cut off that gave a strong indication of FN onset the day after measurement. That is an interesting observation. Were antibiotics used based on an increase of CRP even before FN occured to explain this phenomenum?
  • were any of these parameters associated with a severe course of FN? Such as ICU requirements or death?
  • Daily CRP measurements are not being done in many current institutions. Would the authors recommend that daily CRP measurements become routine assessments? 

Author Response

Reviewer. This is a well written and statistically sound paper that investigated the use of CRP and clinical characteristics as predictive parameters in prolonged febrile neutropenia in transplant patients. The reviewer only has some minor comments/concerns.

Authors. We thank You for the positive comments regarding our work. We have reviewed the manuscript in order to correct eventual minor spell check.

Question 1. Was CRP measured daily? The authors show that the CRP  level of 12 mg/dL is predictive of prolonged FN, however there was no cut off that gave a strong indication of FN onset the day after measurement. That is an interesting observation. Were antibiotics used based on an increase of CRP even before FN occured to explain this phenomenum?

Response 1. CRP was monitored daily in all patients as mentioned in the Materials and Methods section, paragraph “Screening and management of febrile neutropenia”. In our experience the initial CRP values are subsequently influenced by the sensibility to initial treatment response to empiric antibiotics. Therefore, while empiric antibiotics and later changes in antibiotic approach were based on fever persistence, routine CRP monitoring represented a valid aid in evaluating treatment response together with associated symptoms.

Question 2. Were any of these parameters associated with a severe course of FN? Such as ICU requirements or death?

Response 2. One patient passed away during the described period, therefore, the parameters could not have been evaluated for the requested clinical situations.

Question 3. Daily CRP measurements are not being done in many current institutions. Would the authors recommend that daily CRP measurements become routine assessments? 

Response 3. Based on our experience routine daily CRP measurements could be of aid in the management of FN in SCT setting.